# Detection of balance in the elderly under the influence of stress (DEPIE): A cross-sectional study protocol

Membership of GITES[¶], Bernardo Alarcos[1], Belén Díaz-Pulido[2], Olalla Fernández[3], Sara Fernández-Guinea[4], Antonio García Herraiz[1], María del Mar Lendínez-Chica [1]*, Javier Martínez Muela[5], Susana Nunez-Nagy[2], Yolanda Pérez-Martín[2], Isabel Rodríguez-Costa[2], Gloria M. Rubio González[6], Sara Trapero-Asenjo[2]*, Juan R. Velasco[1]

**1** Deparment of Automatic, Polytechnic School, University of Alcalá, Alcalá de Henares, Spain, **2** Department of Nursing and Physiotherapy, Faculty of Medicine and Health Sciences, University of Alcalá, Alcalá de Henares, Spain, **3** 'Los Olmos' Nursing Home, Castile-La Mancha Health Service, Guadalajara, Spain, **4** Department of Experimental Psychology, Cognitive Processes and Speech Therapy, Complutense University, Madrid, Spain, **5** Nursing Homes, Guadalajara, Spain, **6** 'Javier' Center Day, ASISPA, Madrid, Spain

¶ Membership of the Health technology integration research group (GITES) is also provided in the Acknowledgments.
* sara.trapero@uah.es (ST-A); mar.lendinez@uah.es (MDML-C)

## Abstract

Age-related changes increase frailty and vulnerability to stress in older adults, and stress has been linked to poorer balance and fall risk. However, the mechanisms by which emotional stress may contribute to falls remain unclear. This study explores whether: 1) emotional stressors lead to neuromuscular changes that affect postural control in older adults; and 2) technology can assist in fall prevention by detecting increased risk. A cross-sectional, laboratory-based observational study is being conducted in a single session comparing 30 young adults (18–39 years) and 30 older adults (≥65 years), in which participants are exposed to emotional stressors (high-arousal images) and their immediate neuromuscular and balance responses are recorded. The first participant was enrolled on November 27, 2024. All participants complete a sequence of physical tasks under two conditions: viewing low-arousal and high-arousal images from the International Affective Picture System. The physical tasks involve standing up from a chair, walking to a table, transferring water between bottles, returning to the chair, and sitting down. Emotional responses are assessed using heart rate variability, respiratory rate and subjective feelings of unease. Electromyographic signals are analysed using wearable sensors, ground pressure is recorded via pressure sensors and bottle manipulation is tracked with inertial measurement units. Furthermore, balance is evaluated using the Timed Up and Go Test and the Functional Reach Test, administered before the low-arousal condition and after the high-arousal condition. Comprehensive data analysis will provide new

**Data availability statement:** No datasets were generated or analysed during the current study. All relevant data from this study will be made available upon study completion.

**Funding:** This project has been funded by the European Union and the 'Junta de Castilla-La Mancha' (Operational Programme of Castilla-La Mancha 2021-2027) reference SBPLY/21/180501/000257. The funders did not play any role in the study design, data collection and analysis, decision to publish, or preparation of the manuscript.

**Competing interests:** The authors have declared that no competing interests exist.

insights to aid professionals in designing interventions for detecting and preventing fall risk in older adults. The study was approved by the corresponding ethics committee and registered at ClinicalTrials.gov (NCT06682754) on November 21, 2024. It is being conducted at the University of Alcalá, Spain, and is funded by the European Union and the 'Junta de Castilla-La Mancha'. Principal investigators are Dr. Bernardo Alarcos and Dr. Susana Nunez-Nagy (susana.nunez@uah.es). The collected data will be anonymized and shared in an open-access repository, in line with ethical and open science principles.

## Introduction

The number of people over 60 years of age, as well as the proportion they represent within the general population, is increasing [1]. In 2024, people over 60 made up 28.3% of the population in the European Union [2]. According to estimates by the World Health Organization (WHO), this trend will continue to rise over the coming decades, especially in developing countries [1].

Ageing, on a biological level, is associated with the accumulation of molecular and cellular damage that, over time, reduces the overall capacity of the individual [3,4]. Frailty, in turn, refers to the progressive deterioration of physiological systems related to age, causing a decline in capabilities, greater vulnerability to both internal and external stressors and an increased risk of adverse effects such as falls [3,5].

The WHO states that each year approximately 28% to 35% of people over 65 years old suffer a fall [6], and age is one of the main risk factors for falls [7]. Rodríguez-Molinero et al. [8] studied a sample of 772 Spanish individuals over 64 years old for a year to obtain data on the frequency of falls in the elderly population and the associated risk factors. The study results showed incidence rates similar to those reported by the WHO, with approximately one in three people experiencing a fall over the year and one in ten falling repeatedly. Muscle weakness and impaired balance were among the factors associated with the risk of falls [8].

A recent review by Wang et al. [9] on age-related changes in balance highlights the physiological changes that occur during ageing. On the one hand, age is related to changes in the visual, vestibular and proprioceptive sensory systems, which play a crucial role in spatial orientation. Specifically, the visual system provides information about the position and movement of the body relative to the environment. The vestibular system provides information about the position of the head, which is essential for coordinating body movement and stabilising vision. The proprioceptive system enables constant perception of body position in space, helping to control movement without relying on visual information. With ageing, visual information and acuity are affected, vestibular function deteriorates, and proprioceptive abilities decrease. Additionally, ageing causes a gradual decline in cognitive functions such as memory and attention, which also play an important role in postural control and balance. On the other hand, ageing is related to changes in the musculoskeletal system, such

as muscle weakness and loss of bone mass. All these ageing-related changes predispose individuals to falls due to their impact on postural control and balance [9].

Given that frailty implies greater vulnerability to stress factors, several studies have also linked exposure to emotional stress with an increased risk of falls. Fink et al. [10] studied the association between emotional stress and falls in a prospective cohort of men over 65 years old. The results showed that a history of one or more stressful life events in the previous year increased the risk of falls during the following year [10]. However, Fink et al. [10] did not analyse the time interval between the occurrence of stressful life events and the falls, nor did they collect measures of perceived or physiological stress during those life events. Meanwhile, Möller et al. [11] analysed whether sudden emotional stress could trigger falls. They conducted a study with patients over 65 years old diagnosed with hip or pelvic fractures using structured interviews. They asked whether participants had experienced emotions representing emotional stress (anger, sadness, worry, anxiety or stress) on the day of injury or the day before. If the answer was yes, the exact timing of the feeling within the prior 48 hours and its frequency in the past six months were recorded, estimating annual exposure. The results showed an increased risk of falling and subsequent hip or pelvic fracture up to one hour after an episode of emotional stress [11].

However, the mechanism connecting stressful events with falls remains uncertain [10]. It has been hypothesised that the predisposition to falls following emotional stress could be explained by impaired postural control, balance and vision [11]. It has been suggested that in stressful situations, older adults may perform premature gaze shifts that could impair balance control [11]. Akinlosotu et al. [12] researched the influence of mental stress on balance perturbation responses in 23 older adults. Stress was induced through a challenging mental arithmetic task. Perturbations were applied on an instrumented treadmill designed to challenge balance within a safe environment. The results showed that mental stress immediately prior to balance perturbation reduced the effectiveness of grip responses on the handrail to maintain balance. Thus, the authors emphasised the importance of considering mental stress as a possible risk factor for falls and called for further studies with larger samples [12]. More recently, Retamal-Matus et al. [13] explored the relationship between perceived stress, physical performance and balance in 40 people over 60 years old. Perceived stress over the previous month was assessed using the Perceived Stress Scale, physical performance with the Short Physical Performance Battery (SPPB) and balance through a force platform where participants stood for 10 seconds under normal or disturbed conditions with eyes open or closed. No significant correlations were found between perceived stress and physical performance or postural control, respectively. However, as the authors note, the small sample size may have influenced the results, and the relationship between postural balance and stress is complex, requiring further research to better understand the underlying mechanisms [13]. Therefore, further research is needed to study the interactions between cognition, stress and postural control in the context of postural instability and risk of falls in older adults.

Additionally, identifying the risk factors that lead to a fall is crucial to predicting and preventing them. Traditionally, subjective and semi-subjective methods such as questionnaires and functional performance scales have been used to identify fall risk [14]. However, in recent years, different evaluation methods based on advanced technologies have been proposed to assess fall risk more objectively and reliably [14,15]. A recent systematic review by González-Castro et al. [16] aimed to analyse the evidence on the application of Artificial Intelligence (AI) in data analysis related to postural control and risk of fall. The results indicate that wearable devices (accelerometers) and deep learning AI techniques are the most widely used methods for this purpose. They also show that AI applications analysing postural control and fall risk data could be a valuable tool for creating predictive models of fall risk. Therefore, further research is needed on how technology could assist in fall prevention through early risk detection.

For all these reasons, the present project was born, titled 'Use of Devices for Detecting Balance in Older Adults under Stress Influence ("DEPIE" from the Spanish acronym)'. Its aim is to detect whether exposure to stressful situations causes neuromuscular changes that affect postural and motor control in older adults. Studying these questions through different

technological systems also allows exploration of whether technology can help prevention by detecting increased risk of falls.

Specifically, emotional stressors are expected to induce detectable neuromuscular changes via the proposed sensors. It is expected that high-arousal IAPS images will generate autonomic nervous system activation, increasing heart rate variability, possibly respiratory rate, and feelings of unease (measured with a VAS scale from 0 to 10). This may cause alterations in muscle activation, such as increased electrical activity in key muscles (tibialis anterior, gluteus medius, biceps femoris, vastus lateralis, etc.) detected by electromyography; greater sway in ground pressure (Fitness MAT) and displacements with some instability during tasks like standing up, sitting down, walking, or pouring water. In older adults, these effects could exacerbate fall risk, as observed in prior studies with acute perturbations [11,12].

Specifically, the general objectives of the DEPIE project are:

1. To verify that detectable imbalances and sway appear in postural control while standing and during multisensory gross and fine motor tasks when exposed to emotional stressors in both young adults and older adults.

2. To verify whether these imbalances and sway increase the risk of falls in older adults.

3. To verify whether data from sensors can be useful in generating new information through analysis using artificial intelligence and data mining algorithms, assisting health professionals in decision-making.

The specific objectives of the DEPIE project are:

1. To measure the variations in balance and activation of anti-gravity muscles in young and older subjects before, during and after exposure to emotional stressors.

2. To describe how posture in standing and multisensory gross and fine motor functional tasks changes in independently living young and older adults before, during and after exposure to emotional stressors.

3. To compare the variations in balance, mass displacement and muscle activation generated by emotional stressors.

4. To develop a system for data acquisition, processing and storage with a cloud server, edge computing near the sensors and sensors with wireless connectivity.

5. To develop interfaces between commercial sensors and the data acquisition system.

6. To conduct a joint analysis of the data obtained from the sensors and the assessments from health professionals. To apply artificial intelligence and data mining techniques to assess the feasibility of generating information that supports the decision-making of these professionals. Baseline cognitive/physical assessments stratify participants by frailty and cognition, enabling subgroup analyses of stress effects by vulnerability. Conducted once at baseline, TUG/FRT repeat post-intervention measures functional balance changes.

## Materials and methods

### Study design

This is a cross-sectional, laboratory-based observational study conducted in a single session. Outcome assessors and data analysts will be masked to group allocation and experimental condition. Physiotherapists and neuropsychologists carry out the initial assessments and conduct the interventions. Data are stored automatically, and the technologists are responsible for analysing the data obtained.

The schedule for enrolment, intervention and evaluation in the study can be found in Fig 1.

| | STUDY PERIOD | | | | |
|---|---|---|---|---|---|
| | Enrolment | Baseline | Experimental | | Close-out |
| **TIMEPOINT** | $-t_1$ | $0$ | $t_1$ | $t_2$ | $t_3$ |
| **ENROLMENT:** | | | | | |
| Eligibility screen | X | | | | |
| Informed consent | X | | | | |
| **INTERVENTIONS:** | | | | | |
| Baseline test | | | X | | |
| Experimental test | | | | X | |
| **ASSESSMENTS:** | | | | | |
| Inclusion/Exclusion criteria | X | | | | |
| Baseline cognitive assessment (ACE-R, MMSE, TAVEC, ECog, GDS, TMT, Rey Complex Figure Test, VFT, Zoo Map Test) | | X | | | |
| Baseline physical assessment (SPPB, Mini-BESTest, TUG, FRT) | | X | | | |
| Primary outcome variables (EMG variables, pressures on the floor, acceleration and deceleration in object handling) | | | X | X | |
| Primary outcome variables (TUG, FRT) | | X | | | X |
| Secondary outcome variables (HRV, RR, VAS) | | | X | X | |

**Fig 1. SPIRIT diagram.** Schedule for enrolment, intervention, and assessment of the study. ACE-R, Addenbrooke's Cognitive Examination-Revisado; ECog, Everyday Cognition Scale; EMG, electromyography; EVA, Visual Analogue Scale; FRT, Functional Reach Test; GDS, Geriatric Depression Scale de Yesavage; HRV, Heart Rate variability; Mini-BESTest, Mini Balance Evaluation Systems Test; MMSE, Mini Mental State Examination; RR, Respiratory Rate; SPPB, Short Physical Performance Battery; TAVEC, Test de Aprendizaje Verbal España-Complutense; TMT, Trail Making Test; TUG, Timed Up and Go Test; VFT, Verbal Fluency Test.

## Ethical approval and trial registration

The study is conducted in accordance with the recommendations of the Declaration of Helsinki and has been approved by the Ethics Committee for Research and Animal Experimentation of the University of Alcalá (CEIP/2023/5/110). A copy of the protocol submitted to the ethics committee can be found in Supplementary Information S2 File. Furthermore, the study has been registered on ClinicalTrials.gov (NCT06682754).

## Participants

Participants are divided into two groups: young adults (18–39 years) and older adults (≥65 years). Inclusion criteria are: age between 18–39 years for the young group or ≥65 years for the older group; ability to understand spoken and written Spanish sufficiently to follow the study information and instructions; corrected or uncorrected vision sufficient to view the image-based tasks; ability to walk safely over short distances in the laboratory setting, with or without usual assistive devices; and willingness to provide written informed consent. Participants are included if they wish to take part voluntarily and do not present any of the following exclusion criteria: illness, injury or previous trauma that contraindicates muscle exertion, balance exercises and/or walking; physical or mental illness that contraindicates exposure to emotionally stressful stimuli, such as severe depression or psychosis; or difficulty understanding the study information and signing the consent form.

To calculate the sample size, an a priori power analysis was performed. The significance level (α) was set at 0.05, with a statistical power (1 – β) of 0.80, assuming a medium effect size (f = 0.25) to detect the main interaction effect between the group factor (younger vs. older adults), and the stimulus factor (non-stressful vs. stressful) in the context of a 2x2 Mixed ANOVA (F-test, two tails). The selection of a medium effect size is justified because the induction of emotional stress causes a robust and quantifiable functional disruption in older adults' balance reactions, which has been demonstrated in previous studies [12]. G*Power version 3.1.9.7 was used, and it was determined that the required sample size was 27 participants per group. Assuming a potential dropout rate of 10%, the minimum initial sample size is 30 participants for each group.

Young participants are recruited from various schools at the University of Alcalá (UAH), the Complutense University of Madrid (UCM) and other settings such as sports clubs or student associations. Older participants are recruited through contacts at the University for Older Adults at UAH, the Alcalá City Council, associations, residential centres and others.

Participants are recruited by consecutive non-probabilistic sampling and no randomization is necessary, as they will be classified into young or older adults according to age. The first participant was recruited on 27 November 2024. Participant recruitment is currently ongoing and will conclude in July 2026. The study results are expected in December 2026.

Participant information is dissociated from personal data through a coding system: each participant is assigned a code from the outset that identifies them. This unique code is recorded on a list prior to data collection, which is accessible only to the researchers involved in the project. The data collected during the assessments are securely stored under lock and key in the evaluation room. Participants are free to withdraw from the study at any time, without giving any reason and without any adverse consequences to their rights, well-being, or access to care. Concomitant treatments of participants are permitted during the study, as the measurement of variables involves no risk, harm, or discomfort, and will be conducted in a single session only.

## Material and variables

**Baseline cognitive assessment.** Neuropsychological assessment is used to determine participants' cognitive functioning. This evaluation takes place at the beginning of the study (Fig 2).

The presence of cognitive impairment associated with dementia is assessed using the Spanish version of the Addenbrooke's Cognitive Examination-Revised (ACE-R) [17]. The ACE-R is a cognitive screening tool that evaluates five

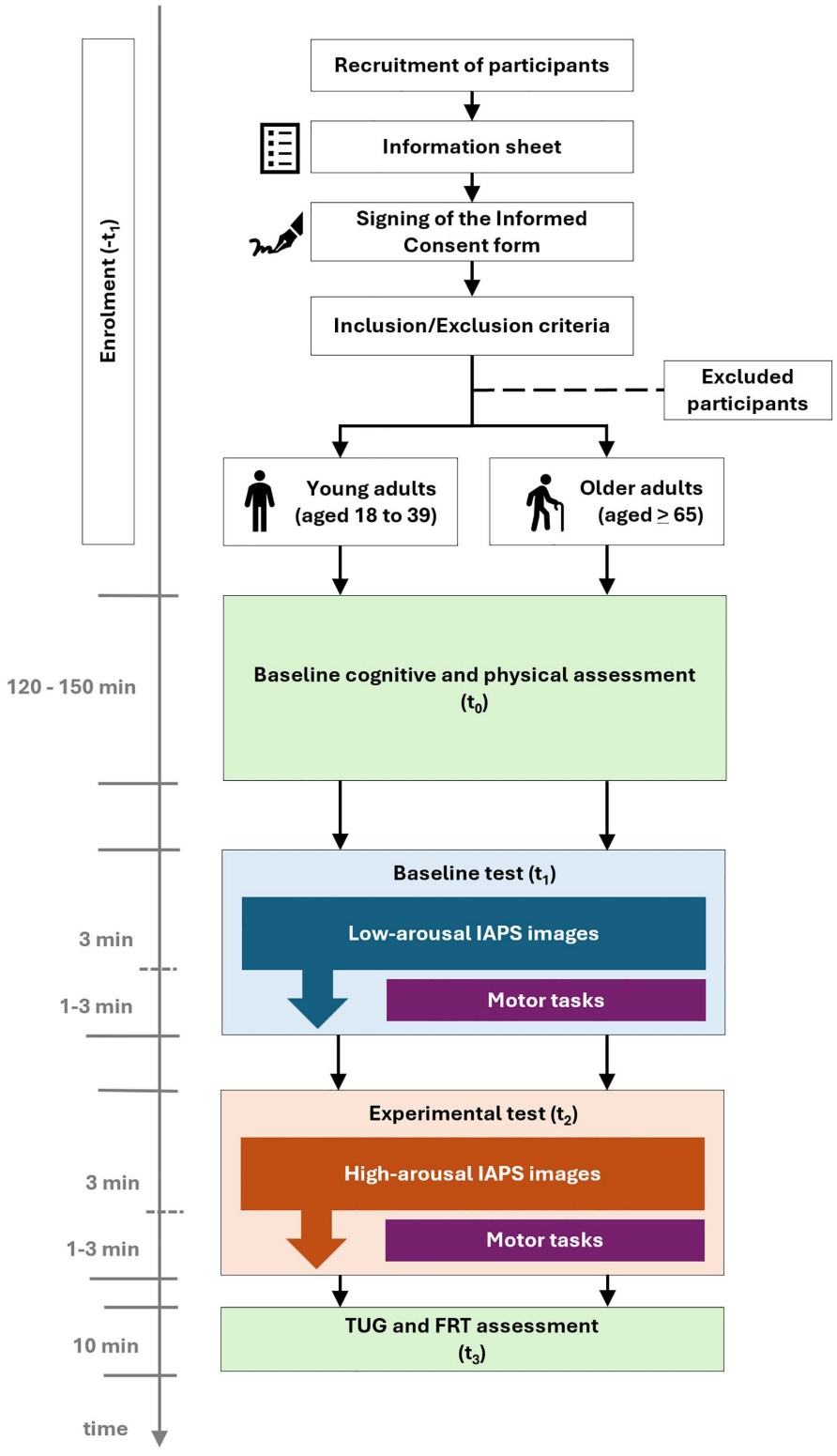

**Fig 2. Study procedure.**

domains. The total score is 100 points, distributed across the following cognitive domains: attention/orientation (18 points), memory (26 points), verbal fluency (14 points), language (26 points) and visuospatial skills (16 points). Higher scores on the ACE-R indicate better cognitive performance. This tool has proven sensitive in detecting cognitive dysfunction in older adults with dementia. Cognitive status is also evaluated using the Mini Mental State Examination (MMSE) [18]. The ACE-R incorporates the MMSE, so an MMSE score can be obtained through administration of the ACE-R. The MMSE totals 30 points, with higher scores indicating better cognitive functioning.

Verbal learning and memory are assessed using the Spanish Verbal Learning Test (TAVEC) [19]. This task involves learning a list of words presented to the participant as though it were a shopping list. The list comprises 16 words from four different semantic categories. Five learning trials are conducted with this list (List A), followed by an interference trial with a new list (List B). List B shares the same format as List A, with 16 words drawn from four categories; two of which overlap with List A and two of which are different. This is followed by short-term free recall of List A and a cued short-term recall, where participants are given the semantic categories as prompts. After a 20-minute interval, a long-term free recall test is administered, followed by a cued long-term recall and, finally, and a recognition test. Participant performance is analysed according to the standardised norms for this test.

Cognitive function in daily life is assessed using the Spanish version of the Everyday Cognition Scale (ECog), which has been adapted and validated for older Spanish-speaking adults [20]. Designed for older adults, this scale evaluates daily functioning in the cognitive domains of memory, planning, language organisation, divided attention and visuospatial and perceptual skills. Each domain consists of several questions, and responses are made in comparison to performance on the same tasks 10 years earlier. Response options are: '1 = no change, the same or better', '2 = sometimes worse than before, but not always', '3 = usually a bit worse than before', '4 = always worse than before'. Participants may also respond 'Don't know or not applicable'. An average score is calculated for each domain, along with a total scale score.

Symptoms of depression are assessed using the shortened Spanish version of the Geriatric Depression Scale by Yesavage (GDS) [21]. This version includes 15 yes/no questions and is specifically designed for older adults. A cut-off score of 5 on the short version is used to indicate the presence of depression.

Visual-motor coordination, visual scanning and alternating attention are assessed using a Spanish-language version of the Trail Making Test, for which equivalence with English versions has been established in Spanish-speaking samples [22], which has two parts. In Part A, the participant is asked to connect 25 randomly distributed numbers on a page in ascending order as quickly as possible. In Part B, the participant must connect 25 numbers and letters in alternating order. The time taken to complete each part and the number of errors made are recorded. Significant time differences in task completion and errors made between both versions, suggest switching and alternating attention problems.

Visuoconstructive abilities and visual memory are assessed using the Rey-Osterrieth Complex Figure Test, applying normative data derived from monolingual Spanish-speaking older adults [23]. This test consists of three phases. In the first, the participant is asked to copy a complex figure. Immediately afterwards, the model is removed, and the participant is asked to reproduce it from memory. A delayed recall is then administered after 20 minutes. This allows for assessment of both immediate and delayed visual recall.

Language is evaluated using the Verbal Fluency Test (TFV) [24], which includes phonological and semantic fluency tasks. Participants are asked to produce as many words as possible in one minute for each of the following phonetic categories: words beginning with F, A and S. For semantic fluency, they are asked to name items from the categories: animals, fruits, items of clothing and kitchen utensils. Both correct answers and mistakes are analysed.

Executive functions, specifically planning ability, are assessed using the Zoo Map Test [25]. This test has two versions, Version 1 and Version 2. In Version 1, participants are asked to plan a route to visit specific locations on a zoo map, following a set of rules (e.g., where to start or finish, or which paths may be used only once). In Version 2, participants perform the same task with the same Version 1 rules, but this time are guided step by step. Planning time, execution time and number of mistakes are analysed in this task.

The battery follows a fixed administration order to avoid interference effects and respect normative protocols: (1) ACE-R, (2) TAVEC, (3) ECog, (4) GDS (during TAVEC 20-min delay if needed), (5) TMT A/B, (6) Rey Complex Figure Test (20-min delay between copy/immediate and delayed recall, during which Verbal Fluency Test may be completed if pending), (7) Zoo Map Test. Estimated total duration: 90–120 minutes, varying by participant performance and including breaks as needed.

To mitigate fatigue in older adults, breaks are offered whenever requested by the participant or observed by the examiner (reduced alertness, motor slowing). If excessive fatigue prevents completion, the session is paused or rescheduled within 1 week maximum.

**Baseline physical assessment.** Physical assessment provides insight into the physical condition of the participants.

Frailty is assessed using the Short Physical Performance Battery (SPPB) whose validity, reliability and reference values have been established in older spanish adults [26]. The SPPB consists of three components that assess balance, gait speed and the ability to stand up from and sit down on a chair five times. Each component is scored out of 4 points, based on the time taken by the participant to complete each task. A total score of less than 10 is considered indicative of a high likelihood of frailty [27].

Balance is evaluated using the Spanish version of the Mini Balance Evaluation Systems Test (Mini-BESTest), which has shown good psychometric properties in community-dwelling older adults [28]. This test includes 14 items that assess anticipatory postural adjustments, postural responses, sensory orientation and dynamic gait stability. Each item is rated on a scale from 0 to 2. Lower scores indicate greater balance impairment [28]. Additionally, the Mini-BESTest includes the Timed Up and Go Test (TUG), conducted both with and without a dual task. The TUG [29] measures the time it takes for the participant to rise from a chair, walk three metres, turn around, return to the chair and sit down again. Three trials are conducted, and times exceeding 12 seconds are interpreted as indicative of high frailty risk [27]. In the dual-task TUG, participants are also asked to count backwards in threes from a specified number before and during the task. A reduction in walking speed of more than 10% between the single and dual-task TUG, or stopping either the walking or the counting, indicates increased frailty. Balance is also assessed using the Functional Reach Test (FRT), for which a Spanish version has demonstrated good validity and reproducibility in older adults [30]. This test measures how far a person can reach forward while maintaining their balance without moving their base of support. Greater reach distances indicate a lower risk of falls.

All these tests are performed at the beginning of the study. In addition, the TUG and FRT are repeated after the intervention (Fig 2).

**Outcome measurements.** The primary outcome measures are:

- Changes in muscle activity [31], monitored through surface electromyography (EMG). EMG signals are recorded using Delsys Trigno system devices (Delsys Inc., Boston, USA), offline mode, 1259 Hz sampling bilaterally on the biceps brachii, multifidus, gluteus medius, biceps femoris, medial gastrocnemius, vastus lateralis and tibialis anterior. Electrodes are placed following SENIAM guidelines [32]. After placement and prior to the experimental session, two maximum voluntary contractions (MVCs) are performed for each muscle, with a six-second rest in between. These MVCs are used for data normalisation in subsequent analyses. Time-based EMG outcome measures include the root mean square (RMS), synchronisation of activation between muscle groups, and the ratio of activation between muscles. Frequency-based measures include the median frequency (MDF). EMG data are collected only during motor task testing, both at baseline ($t_1$) and during the experimental test ($t_2$) for all participants. No EMG recording occurs during baseline physical assessments (SPPB, Mini-BESTest/TUG, FRT), which characterize participant frailty prior to stress exposure (Fig 2).

- Ground pressure is measured using a custom-made Fitness MAT DEV KIT 1.9 mat (80×80 cm) developed for this project (Sensing Tex S.L., Sant Cugat del Vallès, Spain). This mat includes pressure sensors that provide precise information on weight distribution and participant behaviour during weight-shifting tasks [33]. The mat features a sensing area

of 160 × 56 cm and a matrix of 2,240 sensors with a spatial resolution of 20 mm between sensors (80 × 28 matrix), local pressure mapping. Each pressure sensor has a measurement range of 20–10,000 mmHg, with 10% accuracy and a linearity coefficient of 0.99. Data are collected using a 12-bit resolution electronic unit with Bluetooth and USB connectivity. Pressure data are recorded during motor task tests at both baseline ($t_1$) and the experimental test ($t_2$).

- Acceleration and deceleration during object manipulation are measured using WT9011DCL devices (WitMotion ShenZhen Co. Ltd., ShenZhen, China). These triaxial IMUs are attached to the bottles participants must handle during fine motor task tests. Data are recorded during both baseline ($t_1$) and experimental ($t_2$) sessions for all participants.

- Balance, additionally assessed using the TUG and FRT before and after the experimental session.

Secondary outcome measures include: heart rate variability (HRV), respiratory rate (RR) and a visual analogue scale (VAS) assessing the sensation of unease, restlessness or discomfort. The VAS consists of a line marked from 0 to 10, where 0 represents 'completely relaxed' and 10 represents 'the greatest sense of unease, restlessness or discomfort you've ever felt'. Participants are asked to mark the point that best reflects how they feel at that moment. These physiological and subjective secondary measures aim to evaluate the emotional impact of the intervention. HRV and RR are recorded using the Hexoskin Pro vest (Hexoskin Inc., Montreal, Canada). Both variables are analysed during the baseline ($t_1$) and experimental ($t_2$) sessions. All participants complete the VAS at the end of both the baseline ($t_1$) and experimental ($t_2$) sessions, respectively.

All data are stored securely on UAH servers (institutional cloud storage with private folder accessible only to researchers; physical test data kept under lock/key in evaluation room). Data protection fully complies with EU GDPR (Regulation 2016/679) and Spanish LOPDGDD 3/2018: participant data dissociated via unique codes from outset, processed only with explicit consent within UAH's educational/research framework, no transfers except as legally required, and all rights (access, rectification, deletion, etc.) exercisable via protecciondedatos@uah.es. Full details provided in Participant information sheet (Supplementary material S1 File). No cloud/edge computing or transmission to remote locations/outside EU occurs.

**Intervention.** Images from the International Affective Picture System (IAPS) [34] are used during both the baseline ($t_1$) and experimental ($t_2$) sessions to expose participants to non-stressful and emotionally stressful stimuli, respectively. The IAPS is one of the most widely used standardised image sets for inducing emotions in experimental research. It has been shown to reliably elicit measurable changes across the three emotional response systems: subjective-verbal, behavioural and physiological [35]. For the baseline ($t_1$), 40 low-arousal IAPS images have been selected, which collectively produce a mean arousal rating below 2.4 in both young and older adults, based on normative data from previous studies [36]. For the experimental session ($t_2$), 40 high-arousal IAPS images were selected, collectively producing a mean arousal rating above 6.7 in both young and older adults [36].

Potential risks: no physical risks from assessments (supervised by trained professionals). High-arousal IAPS images may cause transient emotional discomfort, resolving shortly after exposure ("the sensation will pass shortly" -participant information sheet). Data security breach risk minimized through pseudonymization (unique codes from outset), local storage (UAH servers/OneDrive private folder, researcher access only; physical data under lock/key), and GDPR compliance (EU 2016/679, LOPDGDD 3/2018) – no identifiable personal data collected).

## Procedure

All study procedures are completed in a single laboratory visit. The study is conducted in the laboratories of the University of Alcalá (UAH), located in the Polytechnic Building and the Nursing and Physiotherapy Building. The researcher first provides verbal explanation of the study, answers all questions, and gives participants the information sheet (attached as Supplementary Information S1 File) to read, take home, and discuss with others as needed. Participants confirm understanding and sign the Informed Consent form only when ready. Firstly, participants read the information sheet and sign the Informed Consent form. Following this, baseline cognitive and physical assessments are carried out (Fig 2).

During the baseline test (t$_1$) and the experimental test (t$_2$), all participants perform the same tasks and wear both the Delsys Trigno system devices and the Hexoskin vest.

The baseline test (t$_1$) is conducted first. Fixed order (low- then high-arousal) mimics real-life stress exposure patterns for ecological validity and avoids carryover effects from randomization. Participants are asked to sit on a stable chair and view low-arousal IAPS images for three minutes. After this period, a signal is given for them to begin the motor task test (gross motor, fine motor and balance tasks). This motor task test involves standing up from the chair, walking 2.3 metres over the pressure mat to a table, emptying the contents of one bottle (water) into another positioned on the table, turning around, walking back to the chair and sitting down again (Fig 3). Low-arousal IAPS images continue to be projected during the motor task test (Fig 2).

Next, the experimental test (t$_2$) is performed. With participants once again seated, they are asked to view high-arousal IAPS images for three minutes. After three minutes, they receive a signal to repeat the same motor task test (gross motor, fine motor and balance tasks) as in the baseline test (t$_1$). During the execution of these motor tasks, high-arousal IAPS images continue to be projected (Fig 2).

Afterwards, the TUG and FRT are repeated. Finally, the Delsys Trigno system devices and the Hexoskin vest are removed.

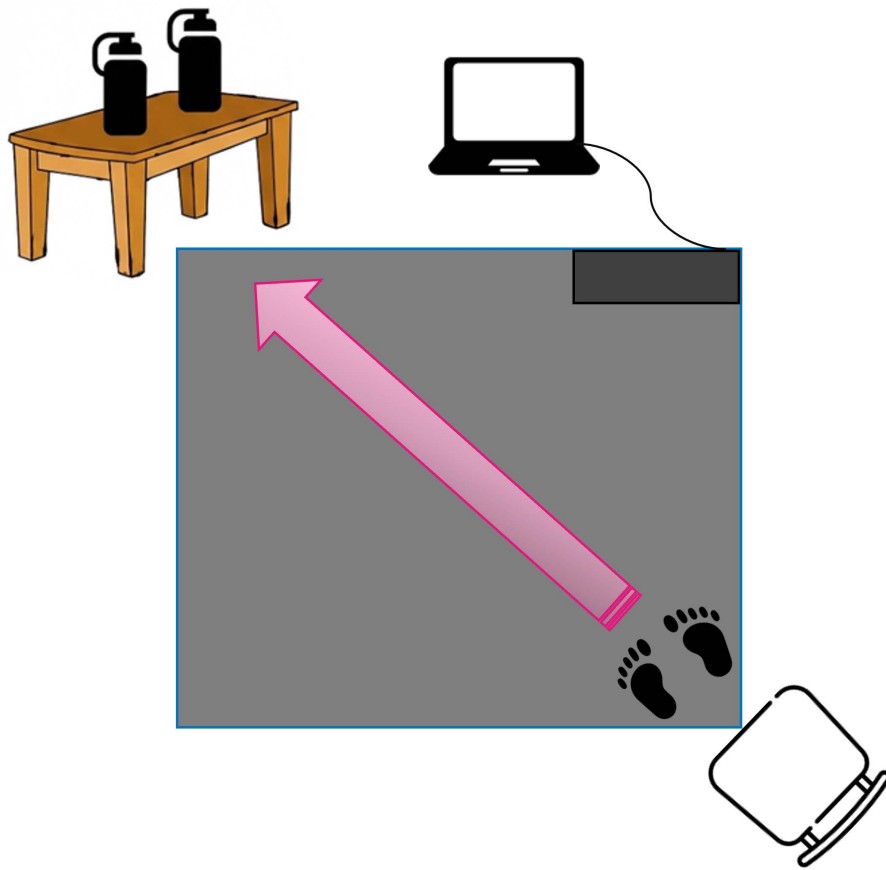

**Fig 3. Diagram of the area for carrying out motor tasks.** By the authors.

### Signal pre-processing

The signals collected from the different sensors will undergo a preprocessing phase, which will include filtering, segmentation and extraction of relevant features. These steps will enable the removal of noise, identification of regions of interest and representation of the signals through meaningful attributes that reflect their dynamic behaviour. The extracted features will serve as the basis for subsequent analysis, aimed at maximising the useful information for classification and clustering tasks.

### Data modelling

One of the main objectives is to verify the potential of sensors to apply machine learning and data mining algorithms. Consequently, the primary outcome variables will be modelled.

Based on the extracted features, different modelling strategies will be explored using machine learning techniques, taking advantage of the functionalities offered by established programming libraries. These analyses will be primarily exploratory, aiming to identify informative patterns and assess the potential of different modelling approaches rather than to develop definitive predictive models. The workflow will follow standard best practices, including data preprocessing, feature scaling, and cross-validation to ensure model robustness and mitigate overfitting. Key performance metrics such as accuracy, precision, recall and F1-score, will be computed to evaluate each model's performance objectively. To prevent data leakage, the dataset will be carefully partitioned into independent training and testing subsets, and transformations such as normalisation and dimensionality reduction, will be applied exclusively within the training pipeline. Overall, this framework supports a transparent and reproducible exploration of dimensionality reduction techniques, supervised and unsupervised learning methods to characterise participants' behavioural patterns and overall condition.

### Statistical analysis

This study also establishes as main objectives the evaluation of postural control alterations under emotional stress in young and older adults (during both quiet standing and motor tasks), as well as determining whether these oscillations increase the risk of falls in the older population. To address these core aims, statistical analyses of the primary and secondary outcome variables will be performed using RStudio version 2024.09.0 (R Core Team).

Initially, a comprehensive descriptive analysis of the sample and the main variables will be carried out to establish their baseline characteristics and verify the quality of the data. Central tendency statistics (mean or median) and dispersion statistics (standard deviation and range) will be calculated for demographic variables. The distribution of the outcome variable data, including psychophysiological stress markers (HRV, RR, and VAS) and functional, biomechanical, and electromyographic metrics (muscle activity, ground pressure, acceleration and deceleration, TUG, and FRT) will be evaluated using the Shapiro-Wilk test to determine their normality. Variables that follow a normal distribution will be presented as mean ± standard deviation, while non-parametric variables will be described using the median and interquartile range. Finally, frequencies and percentages will be used to describe categorical variables.

Patterns of missing data will be examined prior to analysis. When missingness is minimal (e.g., < 5%), analyses will be conducted using complete cases. For variables with more substantial missingness, multiple imputation methods will be applied to reduce potential bias and maintain statistical power.

To assess whether the emotionally stressful stimulus produces psychophysiological effects in participants, secondary outcome variables will first be assessed: HRV, RR, and VAS of unease. Once the effect of stress has been verified, changes in the primary outcome variables will be evaluated: muscle activity (RMS y MDF), ground pressure, acceleration and deceleration, TUG, and FRT. All analyses will be conducted using mixed factorial ANOVAs, assessing the between-subjects factor GROUP (young, older adults) and the within-subjects factor TIMEPOINT (baseline, experimental). Type I error will be set at $\alpha = 5\%$, and partial eta squared ($\eta_p^2$) will be calculated as a measure of effect size. To control

for multiplicity across the different outcome measures, p-values will be adjusted using the Benjamini–Hochberg procedure. Significant results will be subjected to Bonferroni-corrected post hoc *t*-tests, and Cohen's *d* effect sizes will be calculated.

Additionally, subgroup analyses will be conducted within the older adult group, distinguishing between different levels of cognitive impairment and frailty status. The degree of cognitive impairment and frailty status will be determined based on the initial neuropsychological assessment. Correlational analyses between cognitive variables and physical test results will also be carried out.

## Discussion

The purpose of the present project, 'Use of Devices for Detecting Balance in Older Adults Under the Influence of Stress' (DEPIE), is twofold.

On the one hand, the aim of the project is to explore whether exposure to emotional stressors lead to neuromuscular changes that affect postural control in older adults. Previous studies have examined the relationship between stress and balance responses in this population, with contradictory results [12,13]. Akinlosotu et al. [12] showed that acute mental stress immediately prior to a balance perturbation reduced the effectiveness of the balance response. In contrast, Retamal-Matus et al. [13] found no association between perceived stress over the previous month and balance measured on a force platform. However, participants were exposed to different types of stress in each study. Firstly, Akinlosotu et al. [12] induced acute stress via an arithmetic mental task, whereas Retamal-Matus et al. [13] assessed self-reported perceived stress. While acutely induced stress from a mental task is an immediate response to a specific event [37], perceived stress measures the extent to which a person considers life situations to be stressful [38], and this perceived stress may be related to various factors. Additionally, the time interval between stress exposure and the balance task assessment differed between the two studies. Therefore, the detrimental effects reported by Akinlosotu et al. [12] but not observed by Retamal-Matus et al. [13] may be due to the acute nature of the stressor and the short delay between stress exposure and balance assessment. This highlights the need to further investigate the mechanisms by which acute stressors may predispose older adults to falls. Furthermore, among the many stressors encountered in everyday life, previous studies have linked emotional stress to an increased risk of falls in this population [10,11]. However, emotionally stressful events have typically been assessed using retrospective questionnaires and interviews [10,11]. The DEPIE project arises from the need to evaluate the immediate effects of emotional stressors in older adults. Analysing neuromuscular changes in such situations and their effects on postural control and immediate balance will be key to understanding the mechanisms by which stress exposure increases fall risk. This study will contribute to updating knowledge that can support professionals in designing fall prevention interventions. On the other hand, the DEPIE project also aims to explore whether technology can assist in fall prevention by detecting increases in risk.

In recent years, publications on Artificial Intelligence (AI) and falls in older adults have increased significantly [15]. Previous studies have developed AI systems, sensors and communication networks to detect falls as they occur [39,40], and have demonstrated that deep learning can identify falls with high accuracy [41]. Other studies have explored the use of AI techniques to assess fall risk and identify risk factors. This is an area of great interest, given the need for fall risk assessment systems which are more objective and reliable than those traditionally used [14]. In this regard, the systematic review by Olson et al. [42] highlighted that wearable technology using inertial measurement units (IMUs) is highly promising in the detection of changes in gait and balance related to increased fall risk. Similarly, the use of AI, particularly deep learning techniques, also holds promise for fall prevention through the development of predictive models [16]. Despite the promising results so far, this study is needed to delve deeper into the use of technology for detecting increased fall risk. The proposed strategy involves integrating distributed sensors with feature extraction from the recorded signals, with the aim of developing a model capable of identifying postural risk situations during everyday activities. This approach is based on a multisensory architecture capable of simultaneously capturing motion and physiological response variables via devices located both in the environment and superficially on the subject. A system of this kind provides a

more comprehensive understanding of the individual's behaviour and condition in everyday contexts, thus facilitating the identification of potentially unstable situations.

## Conclusion

DEPIE's comprehensive multi-sensor approach (EMG from 7 muscles, pressure mapping, IMUs, HRV/RR) and standardized IAPS stress induction will establish whether acute emotional stress triggers detectable neuromuscular changes affecting postural control, providing foundational data for technology-assisted fall risk assessment systems. This would facilitate early support, promote self-awareness and aid in stress management. The integration of different technologies into the tests conducted during the study will improve the objectivity and accuracy of the assessment. The large-scale acquisition, analysis and processing of data will enable the practical application of the results by a range of professionals in the health and social sciences, not only those working with older adults, but also with people of all ages, as a comprehensive methodology is proposed for the evaluation of balance, fine motor and gross motor tasks. For all these reasons, the development of the DEPIE project will represent a significant contribution to the state of the art.

## Supporting information

**S1 File. The SPIRIT 2025 checklist.**
(DOCX)

**S2 File. The copy of the protocol approved by the ethics committee.**
(DOCX)

## Acknowledgments

Membership of the Health Technology Integration Research Group (GITES) in alphabetical order: Bernardo Alarcos, Belén Díaz-Pulido, Olalla Fernández, Sara Fernández-Guinea, Antonio García Herraiz, María del Mar Lendínez-Chica, Javier Martínez Muela, Susana Nunez-Nagy, Yolanda Pérez-Martín, Isabel Rodríguez-Costa, Gloria M. Rubio González, Sara Trapero-Asenjo, Juan R. Velasco.

The Principal Investigators for the project are Dr. Susana Núñez Nagy (University of Alcalá) and Dr. Bernardo Alarcos (University of Alcalá). The lead author of the group is Dra. Susana Núñez Nagy, whose contact email is: susana.nunez@uah.es

## Author contributions

**Conceptualization:** Bernardo Alarcos, Belén Díaz-Pulido, Olalla Fernández, Antonio García Herraiz, Sara Fernández-Guinea, María del Mar Lendínez-Chica, Javier Martínez Muela, Susana Nunez-Nagy, Yolanda Pérez-Martín, Isabel Rodríguez-Costa, Gloria M. Rubio González, Sara Trapero-Asenjo, Juan R. Velasco.

**Data curation:** Bernardo Alarcos, Antonio García Herraiz, María del Mar Lendínez-Chica, Juan R. Velasco.

**Formal analysis:** Bernardo Alarcos, Antonio García Herraiz, María del Mar Lendínez-Chica, Juan R. Velasco.

**Funding acquisition:** Bernardo Alarcos, Susana Nunez-Nagy.

**Investigation:** Belén Díaz-Pulido, Olalla Fernández, Sara Fernández-Guinea, Javier Martínez Muela, Susana Nunez-Nagy, Yolanda Pérez-Martín, Isabel Rodríguez-Costa, Gloria M. Rubio González, Sara Trapero-Asenjo.

**Methodology:** Bernardo Alarcos, Belén Díaz-Pulido, Olalla Fernández, Antonio García Herraiz, Sara Fernández-Guinea, María del Mar Lendínez-Chica, Javier Martínez Muela, Susana Nunez-Nagy, Yolanda Pérez-Martín, Isabel Rodríguez-Costa, Gloria M. Rubio González, Sara Trapero-Asenjo, Juan R. Velasco.

**Project administration:** Bernardo Alarcos, Susana Nunez-Nagy.

**Resources:** Bernardo Alarcos, Belén Díaz-Pulido, Olalla Fernández, Antonio García Herraiz, Sara Fernández-Guinea, María del Mar Lendínez-Chica, Javier Martínez Muela, Susana Nunez-Nagy, Yolanda Pérez-Martín, Isabel Rodríguez-Costa, Gloria M. Rubio González, Sara Trapero-Asenjo, Juan R. Velasco.

**Software:** Bernardo Alarcos, Antonio García Herraiz, María del Mar Lendínez-Chica, Juan R. Velasco.

**Supervision:** Bernardo Alarcos, Susana Nunez-Nagy.

**Validation:** Bernardo Alarcos, Antonio García Herraiz, María del Mar Lendínez-Chica, Sara Trapero-Asenjo, Juan R. Velasco.

**Visualization:** María del Mar Lendínez-Chica, Susana Nunez-Nagy, Sara Trapero-Asenjo.

**Writing – original draft:** Bernardo Alarcos, Belén Díaz-Pulido, Sara Fernández-Guinea, María del Mar Lendínez-Chica, Susana Nunez-Nagy, Yolanda Pérez-Martín, Sara Trapero-Asenjo.

**Writing – review & editing:** Bernardo Alarcos, Belén Díaz-Pulido, Olalla Fernández, Antonio García Herraiz, Sara Fernández-Guinea, María del Mar Lendínez-Chica, Javier Martínez Muela, Susana Nunez-Nagy, Yolanda Pérez-Martín, Isabel Rodríguez-Costa, Gloria M. Rubio González, Sara Trapero-Asenjo, Juan R. Velasco.

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
