## [Decision Letter · Decision Letter 0]

12 Dec 2025

PONE-D-25-43372Detection of balance in the elderly under the influence of stress (DEPIE): a cross-sectional study protocolPLOS One

Dear Dr. Trapero-Asenjo,

Thank you for submitting your manuscript to PLOS ONE. After careful consideration, we feel that it has merit but does not fully meet PLOS ONE’s publication criteria as it currently stands. Therefore, we invite you to submit a revised version of the manuscript that addresses the points raised during the review process.

We look forward to receiving your revised manuscript.

Kind regards,

Shu Morioka, Ph.D.

Academic Editor

PLOS One

**Journal Requirements:**

1. When submitting your revision, we need you to address these additional requirements. Please ensure that your manuscript meets PLOS ONE's style requirements, including those for file naming. The PLOS ONE style templates can be found at https://journals.plos.org/plosone/s/file?id=wjVg/PLOSOne_formatting_sample_main_body.pdf and https://journals.plos.org/plosone/s/file?id=ba62/PLOSOne_formatting_sample_title_authors_affiliations.pdf 2. Please amend your list of authors on the manuscript to ensure that each author is linked to an affiliation. Authors’ affiliations should reflect the institution where the work was done (if authors moved subsequently, you can also list the new affiliation stating “current affiliation:….” as necessary). 3. One of the noted authors is a group or consortium [Membership of the Health Technology Integration Research Group (GITES)]. In addition to naming the author group, please list the individual authors and affiliations within this group in the acknowledgments section of your manuscript. Please also indicate clearly a lead author for this group along with a contact email address. 4. If the reviewer comments include a recommendation to cite specific previously published works, please review and evaluate these publications to determine whether they are relevant and should be cited. There is no requirement to cite these works unless the editor has indicated otherwise.

**Additional Editor Comments:**

Dear Authors,

The reviewers recognise the importance of your research question and highlight several strengths: a clear rationale, comprehensive multi-sensor assessment, and an open-science approach with ethics approval and trial registration already in place.

However, all three reviewers identify issues that must be addressed before acceptance. On this basis, the editorial decision is: Major Revision.

Key areas requiring revision:

1. Study design and terminology Revise the description so that terminology is accurate and consistent. In particular, avoid describing the study as "double-blind," as this term applies to interventional designs, not cross-sectional observational studies. Also clarify what "analytical" means in this context, or use more standard terminology. State clearly that this is a cross-sectional, laboratory-based observational study with outcome assessors and/or data analysts masked where appropriate.

2. Primary outcomes, statistical analysis, and sample size Define a single primary outcome (or a clearly prioritised set) and distinguish it from secondary and exploratory outcomes. Align the sample-size calculation with the primary outcome and intended statistical test, and justify the chosen effect size or acknowledge in the limitations that smaller effects may be underpowered. Provide a more structured statistical analysis plan, including handling of missing data and multiplicity control. Clarify the role of machine-learning analyses as exploratory and outline the basic framework (cross-validation, performance metrics, avoidance of data leakage).

3. Inclusion criteria, consent, and risk Specify inclusion criteria beyond age: for example, language comprehension, corrected vision sufficient for image-based tasks, and ability to walk safely. Expand the informed-consent description (verbal explanation, time to consider, opportunity to ask questions). Revisit the risk statement: acknowledge potential transient distress from high-arousal images, specify stopping criteria, and describe debriefing and support options.

4. Data protection and GDPR Clarify how data flow from sensors to storage and analysis (local vs. vendor cloud, encryption, pseudonymisation, retention). Explain briefly how the procedures comply with GDPR, particularly if any data are transmitted or stored outside the EU.

5. Alignment of measures with hypotheses and study flow Link each class of measure to the main hypotheses—a concise table mapping hypotheses to measures would help. Provide a clearer timeline or flow diagram, including approximate durations and the rationale for the fixed order of conditions. Clarify measurement scope (e.g., whether EMG is recorded during TUG and FRT).

6. Conclusion, data availability, and minor issues Reconsider statements in the Conclusion that may overstate what the study can demonstrate (e.g., the claim that results will "make it possible to take action before stressful situations arise"). Ensure the data-availability statement is internally consistent, indicating the planned repository, timing of release, and what will be shared. Correct minor inconsistencies (e.g., reference formatting) in line with journal style.

In your revision, please address all points raised by Reviewers 1–3 in a point-by-point response, indicating where each change has been made. Where you disagree or cannot implement a suggestion, provide a brief justification.

We look forward to receiving your revised manuscript.

Reviewers' comments:

Reviewer's Responses to Questions

**Comments to the Author**

1. Does the manuscript provide a valid rationale for the proposed study, with clearly identified and justified research questions?

Reviewer #1: No

Reviewer #2: Yes

Reviewer #3: Partly

2. Is the protocol technically sound and planned in a manner that will lead to a meaningful outcome and allow testing the stated hypotheses?

Reviewer #1: Partly

Reviewer #2: Yes

Reviewer #3: Partly

3. Is the methodology feasible and described in sufficient detail to allow the work to be replicable?

Reviewer #1: No

Reviewer #2: Yes

Reviewer #3: No

4. Have the authors described where all data underlying the findings will be made available when the study is complete?

Reviewer #1: Yes

Reviewer #2: Yes

Reviewer #3: No

5. Is the manuscript presented in an intelligible fashion and written in standard English?

Reviewer #1: Yes

Reviewer #2: Yes

Reviewer #3: Yes

6. Review Comments to the Author

You may also provide optional suggestions and comments to authors that they might find helpful in planning their study.

**Reviewer #1:**The manuscript requires further improvements particularly on the study design writeup.

Abstract: Double-blind is a feature of interventional designs (e.g., randomized controlled trials, not observational cross-sectional ones. If the study is analytical, it can compare two or more groups (e.g., exposed vs. non-exposed, etc), but it cannot be double-blind if cross sectional.

Line 210: The detailed inclusion criteria is to be provided.

Line 216: More details of the sample size calculation are to be provided, e.g., one or two-tailed test, outcome variable, and effect size figure, test family, statistical test, type of power analysis etc

The study design is unclear. A proper flow chart to illustrate the study design from the beginning, participants, and how the participant is assigned, the exposed groups/intervention groups, duration/period, the assessment point, outcome etc is to be provided. Currently, in the absence of random allocation, the study appears to adopt a prospective cohort design.

Line 270, Line 284, Line 293, Line 304, Line 321, Line 333: The language version of the scales is to be stated.

Line 448-449: The focus is to be on the primary outcome and followed by secondary outcome and not vice versa

**Reviewer #2:** This is a clearly written and scientifically meaningful protocol with strong justification and comprehensive methodology. The integration of objective technologies such as EMG, pressure-sensing systems, IMUs and physiological monitoring is a notable strength, enhancing measurement precision and enabling advanced analytic approaches. The study is feasible and ethically well prepared.

Strengths

Clear rationale addressing gaps in previous research.

Well-defined study procedures and transparent reporting.

Objective and validated outcome measures.

Realistic recruitment plan and open-science data-sharing policy.

Ethical approval and trial registration already in place.

Minor suggestions

Sample-size clarification

The sample-size calculation assumes a medium effect size (f = 0.25) with α = 0.05 and 80% power, leading to 27 participants per group. While reasonable for detecting medium or larger effects in a 2×2 mixed ANOVA framework, smaller interaction effects are common in clinical and rehabilitation research. A brief justification of the chosen effect size based on prior literature, or mention in the limitations that smaller effects may be underpowered, would enhance transparency without requiring changes to the study plan.

Cognitive assessment procedure

The cognitive evaluation consists of several standardized tests, but the manuscript does not indicate the approximate duration required for each assessment or the total expected testing time. Providing estimated administration times would be helpful for evaluating feasibility and participant burden.

Additionally, clarification on whether the order of cognitive tests is fixed or randomized would be useful. In older adults, accumulated fatigue could influence performance and increase variability, and noting any mitigation strategy (e.g., breaks, counterbalancing) would strengthen the protocol description.

Final Statement

Overall, this protocol is thoughtfully developed and addresses an important area in ageing research. The requested clarifications are minor and intended only to improve clarity and reproducibility. I recommend acceptance following minor revisions.

**Reviewer #3:** Review PLOS ONE

PONE-D-25-43372

"Detection of balance in the elderly under the influence of stress (DEPIE): a cross-sectional study protocol"

This is a paper detailing the protocol of a cross sectional laboratory study of younger and older volunteers. The main topic of the study and study protocol is the assessment of the effect of emotion on balance as well as developing a pathway for collecting and analysing data from a variety of sources and sensors. This is related to the clinical problem of falls risk assessment.

The paper describes an extensive battery of assessments of cognition, depression, physical assessment, muscle activity, balance, heart rate and respiratory rate. The cognitive test battery includes at least 7 tests of cognitive functioning with ACE-R with 5 domains, including the MMSE; Spanish Verbal Learning Test, ECog, TMT A and TMTB, Rey complex figure test, Verbal fluency test, Zoo Map test. The physical assessment includes at least 4 tests: the SPPB, Mini-BESTtest with TUG, and the Functional Reach Test. The muscle activity and balance tests are by surface electromyography with sensors on at least 7 different muscle regions, accelerometers on manipulated objects and pressure sensors on the surface the participant is standing on. Heart rate, heart rate variability and respiratory rate are measured by a body worn sensor suite.

I do not have any connections or other relations to the authors, funders or institution and I am competent to review the paper. The paper is well written and the hypothesis is interesting. I think there is insufficient information on how the data will be analysed, the consent procedure and on how data will be handled safely when using commercial sensors. I also think the conclusion is overstating the importance of the study to the clinical problem of falls. I think the paper should be published, as long as the authors are willing to supply more information, and there is no problem with how sensitive data are handled.

Comments related to specific parts of the paper:

Abstract.

What does the word “analytical” signify? . “A cross-sectional, analytical, double-blind study is being conducted comparing 30 young adults (18–39 years) and 30 older adults (≥65 years).”

Page 7 of the pdf document:

“No datasets were generated or analysed during the current study. All relevant data from this study will be made available upon study completion”. Please elaborate. How can you conduct a study without generating data? And please specify how the data will be made available.

Lines 157-183 Please include more information on the suspected effects of stress on the results.

Lines 157-183: I do not understand how the many tests of cognitive and physical ability relate to the research aims. Please explain.

Lines 210-214:I note that being deaf or visually impaired is not a contraindication to participation in the study. How do you intend to handle these participants?

Line 214: Please supply more information on the consent procedure. You state: “Firstly, participants read the information sheet and sign the Informed Consent form. Following this, baseline cognitive and physical assessments are carried out (Fig. 2).” Are the participants verbally informed about the study? Are they able to take the time to consider participation? Can they ask questions? Consult with others?

Line 244, Fig 2. The test procedure is to view the low arousal pictures first, then the high-arousal pictures. Did you consider randomizing the order? Why not? Please explain. Please supply some information about typical time spent in the different parts of the experiment.

Lines 445-460 Do you have a more detailed statistical analysis plan? Is it possible to make it part of the paper? The experiment includes both a data modelling part and an statistical analysis plan, how do these two parts fit together?

Lines 215-219: Sample size calcutions. Please explain this in more detail. What statistical analysis is to be used? What are the estimated averages and variability of the different assessments. Did you do sample size calculations for each assessment separately?

Lines 340-353: Wil the EMG data be collected during physical test? The SPPB, Mini-BESTtest with TUG, and the Functional Reach Test?

Lines 342-367: Is it necessary to include information on software version of the Delsys Trigno, Fitness MAT DEV KIT, WT9011DCL devices and the Hexoskin Pro vest? Can you please describe how data is handled by these commercial devices? Is any data transmitted or passed through remote locations? Are any of these locations outside the European Union? How does the study comply with GDPR?

Lines 396-397: Please describe the potential harm to the participants if sensitive data are compromised. Please also describe the potential emotional harm of viewing disturbing images.

Lines 404-405: Please be specific about what you mean by experiment? Is this the period from the consent form is signed, until the participants leave the laboratory?

Lines 430-444. I am not competent to assess the paragraph on data modelling

Lines 519-520: “The study of all the issues addressed in the DEPIE project will make it possible to take action before stressful situations arise,” This is a strong statement and I do not consider et supported by your study design. Please reconsider the sentence..

Lines 642-643, reference number 27. Please use standard formatting of references. I am not able to assess references in Spanish.

Lines 658-660. Please use standard formatting of references. “Hermens HJ, Freriks B, Merletti R, Stegeman D, Blok J, Rau G, et al. European Recommendations for Surface ElectroMyoGraphy Results of the SENIAM project. Roessingh research and development. 1999; 8(2).” Are you referring to “Hermens, Hermie J., et al. "European recommendations for surface electromyography." Roessingh research and development 8.2 (1999): 13-54.»

End of comments.

7. PLOS authors have the option to publish the peer review history of their article (what does this mean?). If published, this will include your full peer review and any attached files.

Reviewer #1: No

Reviewer #2: **Yes:**Tomoyuki Shiozaki

Reviewer #3: **Yes:**Ole Martin Steihaug

---

## [Author Response · Author response to Decision Letter 1]

22 Dec 2025

Dear Editor and Reviewers,

We are very grateful to the editor and three reviewers who proposed several constructive and helpful suggestions.

Based on their comments, the present modifications should contribute to improving its quality. All major and minor comments have been considered, and we thank you for your careful reading.

Point-by-point responses can be found in the attached document “Rebuttal Letter”.

We remain at your disposal for any questions.

Best regards,

Sara Trapero-Asenjo and María del Mar Lendínez-Chica

---

## [Decision Letter · Decision Letter 1]

13 Jan 2026

Detection of balance in the elderly under the influence of stress (DEPIE): a cross-sectional study protocol

PONE-D-25-43372R1

Dear Dr. Trapero-Asenjo,

We’re pleased to inform you that your manuscript has been judged scientifically suitable for publication and will be formally accepted for publication once it meets all outstanding technical requirements.

Kind regards,

Shu Morioka, Ph.D.

Academic Editor

PLOS One

Additional Editor Comments (optional):

Reviewers' comments:

Reviewer's Responses to Questions

**Comments to the Author**

1. Does the manuscript provide a valid rationale for the proposed study, with clearly identified and justified research questions?

Reviewer #1: Yes

2. Is the protocol technically sound and planned in a manner that will lead to a meaningful outcome and allow testing the stated hypotheses?

Reviewer #1: Partly

3. Is the methodology feasible and described in sufficient detail to allow the work to be replicable?

Reviewer #1: Yes

4. Have the authors described where all data underlying the findings will be made available when the study is complete?

Reviewer #1: Yes

5. Is the manuscript presented in an intelligible fashion and written in standard English?

Reviewer #1: Yes

6. Review Comments to the Author

You may also provide optional suggestions and comments to authors that they might find helpful in planning their study.

Reviewer #1: The authors have adequately addressed all comments, and I have no further remarks. The manuscript is acceptable for publication.

7. PLOS authors have the option to publish the peer review history of their article (what does this mean?). If published, this will include your full peer review and any attached files.

Reviewer #1: No

---

## [Editor Report · Acceptance letter]

PONE-D-25-43372R1

PLOS One

Dear Dr. Trapero-Asenjo,

I'm pleased to inform you that your manuscript has been deemed suitable for publication in PLOS One. Congratulations! Your manuscript is now being handed over to our production team.

Kind regards,

on behalf of

Professor Shu Morioka

Academic Editor

PLOS One